# Supercomputer framework for reverse engineering firing patterns of neuron populations to identify their synaptic inputs

Matthieu K Chardon[1,2]*, Y Curtis Wang[3], Marta Garcia[4], Emre Besler[5], J Andrew Beauchamp[6], Michael D'Mello[7], Randall K Powers[8], Charles J Heckman[1,9,10]

[1]Department of Neuroscience, Northwestern University, Chicago, United States; [2]Northwestern-Argonne Institute of Science and Engineering (NAISE), Evanston, United States; [3]Department of Electrical and Computer Engineering, California State University, Los Angeles, Los Angeles, United States; [4]Argonne Leadership Computing Facility, Argonne National Laboratory, Lemont, United States; [5]Department of Electrical Engineering, Northwestern University, Evanston, United States; [6]Department of Biomedical Engineering, Northwestern University, Chicago, United States; [7]Intel Corporation, Santa Clara, United States; [8]Department of Physiology and Biophysics, University of Washington, Seattle, United States; [9]Physical Medicine and Rehabilitation, Shirley Ryan Ability Lab, Chicago, United States; [10]Physical Therapy and Human Movement Sciences, Northwestern University, Chicago, United States

*For correspondence: matthieuchardon@gmail.com

**Competing interest:** The authors declare that no competing interests exist.

**Abstract** In this study, we develop new reverse engineering (RE) techniques to identify the organization of the synaptic inputs generating firing patterns of populations of neurons. We tested these techniques in silico to allow rigorous evaluation of their effectiveness, using remarkably extensive parameter searches enabled by massively-parallel computation on supercomputers. We chose spinal motoneurons as our target neural system, since motoneurons process all motor commands and have well-established input-output properties. One set of simulated motoneurons was driven by 300,000+ simulated combinations of excitatory, inhibitory, and neuromodulatory inputs. Our goal was to determine if these firing patterns had sufficient information to allow RE identification of the input combinations. Like other neural systems, the motoneuron input-output system is likely non-unique. This non-uniqueness could potentially limit this RE approach, as many input combinations can produce similar outputs. However, our simulations revealed that firing patterns contained sufficient information to sharply restrict the solution space. Thus, our RE approach successfully generated estimates of the actual simulated patterns of excitation, inhibition, and neuromodulation, with variances accounted for ranging from 75–90%. It was striking that nonlinearities induced in firing patterns by the neuromodulation inputs did not impede RE, but instead generated distinctive features in firing patterns that aided RE. These simulations demonstrate the potential of this form of RE analysis. It is likely that the ever-increasing capacity of supercomputers will allow increasingly accurate RE of neuron inputs from their firing patterns from many neural systems.

## eLife assessment

The study by Chardon et al. is **fundamental** to advancing our understanding of presynaptic control of motor neuron output. Large-scale computer simulations were performed using well-established

single motor neuron models to provide **compelling** evidence regarding the time-varying patterns of inputs that control motor neuron ensembles. The work will interest the community of motor control, motor unit physiology, neural engineering, and computational neuroscience.

## Introduction

Array electrodes that allow simultaneous recording of firing patterns of populations of neurons in mammals have transformed their understanding of the computations implemented by neural networks during natural behaviors and the utilization of these computations for brain-machine interfaces (*Buonomano and Maass, 2009*; *Chandrasekaran et al., 2021*; *Stevenson et al., 2012*; *Stevenson and Kording, 2011*; *Vyas et al., 2020*). However, to understand how a given neural network generates the firing patterns that implement these computations, it is necessary to understand the transformation of inputs to outputs by the constituent neurons. The net effect of ionotropic excitatory inputs in activating a neural circuit is likely to be strongly impacted by the pattern of inhibitory inputs arising from local interneurons and the effects of neuromodulatory inputs acting on G-protein coupled receptors (*Binder et al., 2020*; *Goaillard and Marder, 2021*; *McCormick and Nusbaum, 2014*). In-depth analyses of the resulting complex interactions require intracellular electrode techniques, but these are not yet feasible simultaneously in many neurons during a natural behavior. An alternative is RE the firing patterns recorded by arrays to identify the neurons' inputs and properties.

Thus far, RE has been used to achieve several different goals, including the estimation of circuit structures within simulations with simple neuron models (*Lim et al., 2011*; *Pisokas, 2020*; *Rostro-Gonzalez et al., 2012*) and estimation of local field potentials from measurements of spiking patterns (*Telenczuk et al., 2020*; *Yochum et al., 2019*). In this study, our goal was to harness the computational power provided by the implementation of realistic neuron models on a supercomputer to investigate the feasibility of reverse engineering these model firing patterns to identify the underlying organization of their simulated excitatory, inhibitory, and neuromodulatory inputs. Successful RE of firing patterns into input organization would greatly advance the effectiveness of array recordings in identifying how neural circuits implement their computations.

RE of neural outputs into inputs faces a formidable barrier. Multiple neural systems have been shown to exhibit non-uniqueness, which is manifest as closely similar outputs being produced by many combinations of parameters specifying neuron properties and inputs (*Edelman and Gally, 2001*; *Marder, 2012*; *Prinz et al., 2004*). This set of parameters forms the 'solution space' for that neural system (*Prinz, 2010*). Although likely advantageous for stability and resilience of neurons and circuits (*Goaillard and Marder, 2021*), a large solution space that contains many combinations of excitation, inhibition, and neuromodulation may limit the effectiveness of RE. In addition, the parameters within the solution space may interact in complex, nonlinear ways (*Mukunda and Narayanan, 2017*). Neuromodulatory inputs, for example, may have highly nonlinear effects on how neurons process their ionotropic inputs (*Binder et al., 2020*; *Marder, 2012*; *McCormick and Nusbaum, 2014*). On the other hand, when neuronal output behaviors are complex, non-uniqueness may be reduced by the constraints of matching multiple outputs (*Mukunda and Narayanan, 2017*; *Yang et al., 2022*).

To reduce the impact of non-uniqueness, we chose a well-understood neural system for evaluation of RE, the motoneurons that transform all motor commands into signals for control of muscles (*Heckman and Enoka, 2012*). Motoneuron properties and inputs have been extensively studied using in situ voltage clamp methods in animal preparations, with a focus on neuromodulatory actions (*Heckman and Enoka, 2012*; *Henneman and Mendell, 1981*). This data set has enabled us to generate highly realistic motoneuron models (*Powers et al., 2012*; *Powers and Heckman, 2017*), which we have successfully implemented on a supercomputer for the present study. Furthermore, recent advances in flexible array electrode methods now allow measurement of firing patterns of populations of motoneurons in humans, due to the one-to-one relation between motoneuron action potentials and those of its innervated muscle fibers (*Holobar et al., 2010*; *Hug et al., 2021*; *Negro et al., 2016*). Consequently, RE of these population firing patterns may generate deep insights into the synaptic organization of motor commands in humans as well as other animals.

We used an ensemble modeling approach (*Prinz, 2010*) to determine how non-uniqueness and nonlinearity affected RE identification of the pattern of excitatory, inhibitory, and neuromodulatory inputs that produce motor output. All studies were carried out in silico, so that both inputs and

outputs were known and thus the effectiveness of RE could be accurately quantified. The supercomputer implementation of motoneuron models allowed us to thoroughly explore this input solution space, via more than 300,000 simulations of the effects of many input combinations on firing patterns. Our previous intracellular studies of motoneuron input-output functions suggest that motoneuron firing patterns contain substantial information about their inputs (*Hyngstrom et al., 2008*; *Kuo et al., 2003*; *Lee and Heckman, 1999*; *Lee and Heckman, 1998a*; *Lee and Heckman, 1998b*). We, therefore, hypothesized that RE of motoneuron firing patterns would effectively deal with the problems of non-uniqueness and would, therefore, identify a reasonably small solution space of input combinations. To assess the importance of information contained within neuron firing patterns for achieving a small solution space, we compared the RE of motoneuron firing patterns to the RE of the cumulative spike train (CST), an overall measure of motor output generated by summing all neuron firing patterns together, which has been shown to closely replicate the muscle electromyogram (EMG), which is directly related to muscle force. Our simulation results supported the hypothesis and demonstrated a much smaller solution space for RE of firing of neuron populations than for the CST.

## Results
### Performance of the motoneuron models for integration of inputs
All simulations are run using a set of 20 model motoneurons designed to closely recreate behaviors documented in our extensive database of current and voltage clamp studies in motoneurons within animal preparations. These models were first developed in NEURON (*Hines and Carnevale, 1997*) for a previous study (*Powers and Heckman, 2017*) and have been implemented on a supercomputer for the present work (see Methods). Our goal of investigating the feasibility of RE of motoneuron firing patterns depend on the accuracy of these motoneuron models. *Figure 1* illustrates the performance of our motoneuron models in realistically capturing the integration of excitatory (*Figure 1A*), neuromodulatory (*Figure 1B*) and inhibitory (*Figure 1C*) inputs. Each of these simulated behaviors are considered in the following subsections.

### Common input structure, differences in motoneuron properties
A fundamental point in achieving accurate modeling of the outputs of a pool of motoneurons innervating a muscle is that they are subject to 'common drive' (*De Luca and Erim, 1994*; *Farina and Negro, 2015*; *Heckman and Enoka, 2012*). As a result, motoneuron firing patterns for a specific task exhibit an overall similarity, which is likely essential for efficient force control (*Farina and Negro, 2015*). This point is important for RE, as common drive precludes independent inputs to each motoneuron. Firing patterns, however, are not identical. As in any biological system, noise is present, though this is relatively low in motoneurons (coefficient of variation during steady firing at about 10–20% *Heckman and Enoka, 2012*). We add noise to our excitatory synaptic conductances to match this level, as illustrated in *Figure 1A1*, which illustrates firing patterns of 5 of the 20 motoneuron models in response to a linearly rising and falling excitatory conductance. This figure also shows systematic differences in activation (i.e. recruitment) thresholds. These differences are due to systematic differences in intrinsic electrical thresholds for spiking of motoneurons, which generates an orderly activation (i.e. recruitment) sequence known as Henneman's size principle (*Henneman and Mendell, 1981*). The size principle is essential for the effective utilization of the slow (S) vs fast (F) muscle fibers, with motoneurons innervating slow fibers having the lowest thresholds. S units tend to reach higher firing rates than F units, which require more input to reach their higher thresholds so less is available for rate modulation (*Beauchamp et al., 2022*; *De Luca and Contessa, 2012*; *Hu et al., 2013*).

### Distribution of excitatory input to S vs F motoneurons
In addition to the pattern and amplitude of excitatory input, there is another important aspect of excitatory motor commands, which is that all excitatory inputs studied thus far project differentially to S vs. F motoneurons (*Powers and Binder, 2001*) in contrast, inhibition tends to be uniform (*Powers and Binder, 2001*; *Lindsay and Binder, 1991*) as does neuromodulation (*Lee and Heckman, 1998a*; *Lee and Heckman, 1999*). Descending inputs tend to generate larger synaptic currents in the F's, especially the corticospinal input (*Powers and Binder, 2001*) (see Discussion). As yet only one input favors S motoneurons, but it is an essential one, monosynaptic Ia excitation from muscle spindles

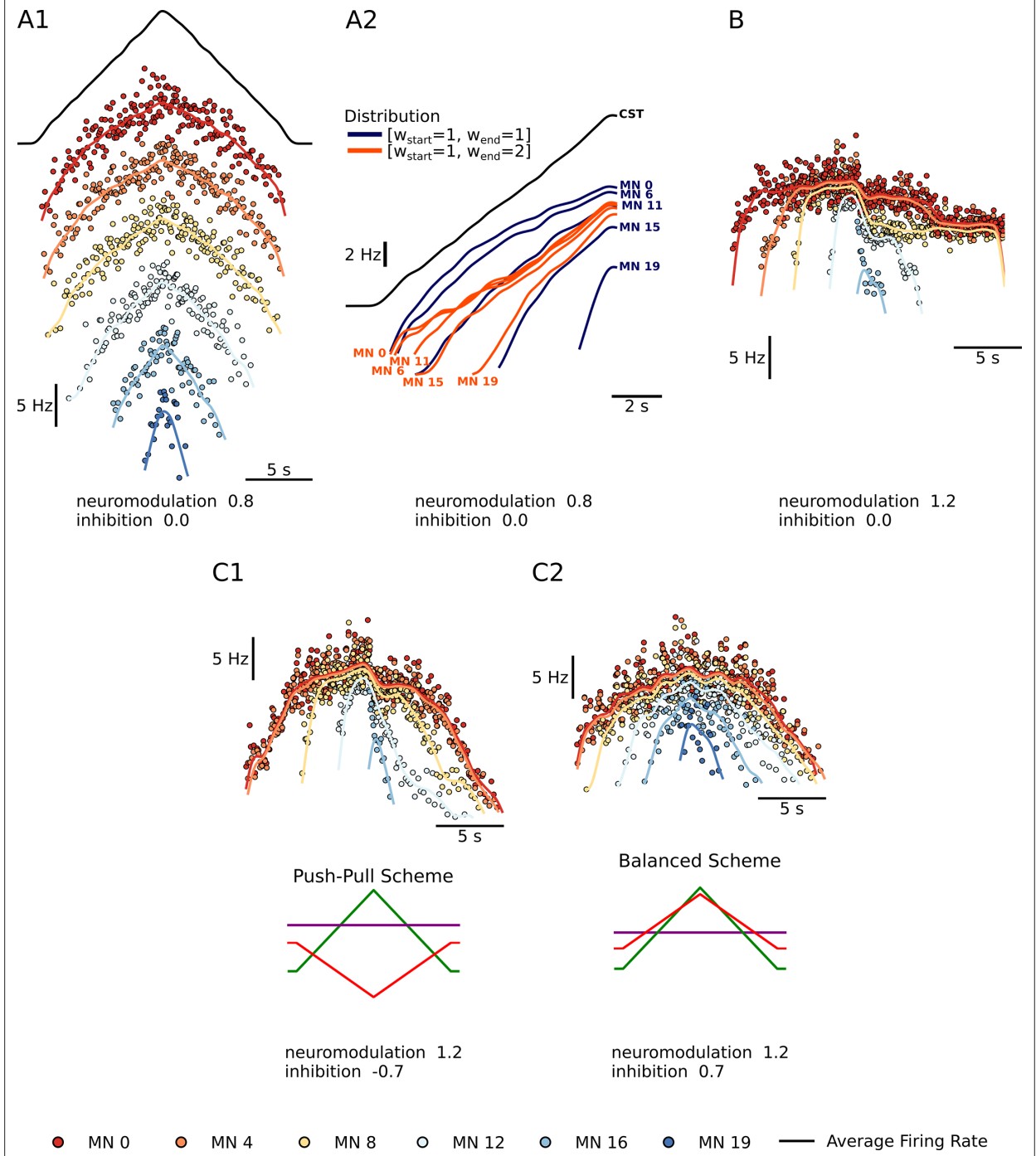

**Figure 1.** Summary plots showing the performance of the motoneuron pool model with respect to the integration of inputs. (**A1**) - Shows a subset of 6 of the 20 motoneuron (MN) in the pool given the same excitatory triangular input, the black trace above. The MN pool follows an orderly activation known as the Henneman's size principle, showing that this MN model can transform a 'common drive' into different MN activities (see section Common input structure, differences in motoneuron properties). (**A2**) - Shows the model's ability to favor S-type vs F-type motoneuron within the pool. The orange $[w_{start}, w_{end}] = [1, 2]$ Distribution favors the fast (F-type) motoneuron. The recruitment of theses F-type motoneurons (e.g. MN 11, MN 15, and MN 19) happens sooner and their maximum firing rates are higher with respect to their $[w_{start}, w_{end}] = [1, 1]$ counterparts. For the slow (S-type) motoneurons the recruitment is the same but their max firing rates decrease with respect to their $[w_{start}, w_{end}] = [1, 1]$ counterparts (see section Distribution of excitatory input to S vs F motoneurons). (**B**) - Shows our model's ability to trigger PIC-type behavior. For a neuromodulation level of 1.2 (max of range tested) and no inhibition, the motoneurons firing rates show a fast rise time, then attenuation, followed by sustained firing (see section Neuromodulation and excitatory input). (**C1 and C2**) - Shows the model's ability to modulate the effects of neuromodulation with inhibition. (**C1**) shows an inhibition in the 'Push-Pull' or reciprocal configuration. (**C2**) shows inhibition in the balanced configurations (see section Neuromodulation and inhibition).

(*Heckman and Binder, 1988*). These differences, though, do not provide independent control of F motoneurons, much less of individual motoneurons, as common drive still dominates and recruitment still follows the size principle (*Heckman and Binder, 1993*). *Figure 1A2*, however, shows that a greater relative excitatory synaptic current in high threshold F motoneurons does tend to compress the range of recruitment thresholds and increase the steepness of rate modulation as input levels rise. This compression/steepening may thus provide RE sufficient information to identify the distribution of excitatory input to S vs F motoneurons and provide information on the relative roles of descending vs sensory input.

## Neuromodulation and excitatory input

We chose to focus on just one type of neuromodulation (see Methods), the monoaminergic input originating in the brainstem, because this system has proved to have particularly powerful effects on motoneuron excitability (*Heckman et al., 2005*; *Heckman and Enoka, 2012*). These monoaminergic axons form a dense and monosynaptic projection onto motoneurons throughout the spinal cord, with the synapses releasing either 5HT (axons originating in the caudal raphe nuclei) or NE (axons from the locus coeruleus) (*Alvarez et al., 1998*; *Holstege and Kuypers, 1987*; *Maratta et al., 2015*). Both 5HT and NE have multiple effects on motoneuron ion channels, but their strongest action is via the facilitation of persistent inward currents (PICs) mediated by dendritic Ca channels (likely CaV 1.3) (*Binder et al., 2020*). Our motoneuron models have been fine tuned to accurately replicate the highly nonlinear interactions between PICs and excitatory inputs (*Powers and Heckman, 2017*). *Figure 1B* shows that, when brainstem neuromodulatory are moderately high and dendritic PICs become strong, three marked nonlinearities become apparent: an initial acceleration (due to PIC activation and consequent amplification of input), followed by attenuation (due to PIC depolarization of dendritic regions, which reduces excitatory driving force) and then hysteresis (offset at a lower level than onset, due to the prolongation of input). In this example, where a pure excitatory drive is applied with no background of inhibition, low threshold motoneurons continue firing long after the input returns to baseline (this is known as self-sustained firing). These nonlinearities are striking in comparison to the nearly linear behavior in the low neuromodulatory state (*Figure 1A1*).

## Neuromodulation and inhibition

Our models also accurately capture the very strong interaction of the PIC with inhibition. Even a small background of inhibition reduces PIC amplitudes (*Hyngstrom et al., 2007*; *Kuo et al., 2003*). *Figure 1C* shows that modulation of inhibitory input that is either reciprocally or proportionally organized with respect to excitation also dramatically alters the pattern of rate modulation. Reciprocal inhibition (*Figure 1C1*) tends to allow strong PIC expression with acceleration, attenuation and hysteresis, whereas proportional inhibition (*Figure 1C2*) tends to flatten rate modulation and reduce hysteresis in fact, of all the patterns shown in *Figure 1*, C1 is closest to that seen in many human muscles (*Beauchamp et al., 2022*; *Heckman and Enoka, 2012*). Thus the nonlinearities in firing patterns are not a straightforward reflection of neuromodulatory input but instead result from a complex interaction between neuromodulation and inhibition.

Given the above effects of the organization of excitation, inhibition, and neuromodulation on firing patterns, our RE effort targeted identification of the following aspects of excitation, inhibition and neuromodulation: amplitude and time course of excitation, distribution of excitation to S vs F motoneurons, amplitude, and time course of inhibition (including the baseline level and then whether inhibition was proportional, constant, or balanced) and level of neuromodulation from low to high.

## Ensemble modeling for reverse engineering

The concept for the RE approach we used is illustrated in *Figure 2*. The left portion shows the flow of motor commands from the brain, brainstem, and spinal cord to motoneurons, which then activates a muscle. The muscle produces an output force. This force is due to the summed actions of recruited motor units, whose spike trains sum to produce an overall electrical signal (EMG). Our focus is on the spike trains themselves, which can be measured in both animals and humans. The pattern of average firing rate is obtained by summing the rates of all active units (see Methods) and is thus referred to as the CST (righthand side, bottom). We then apply RE methods to both the CST and to individual motor unit firing patterns (right, middle), which, if successful, will allow us to estimate the temporal patterns

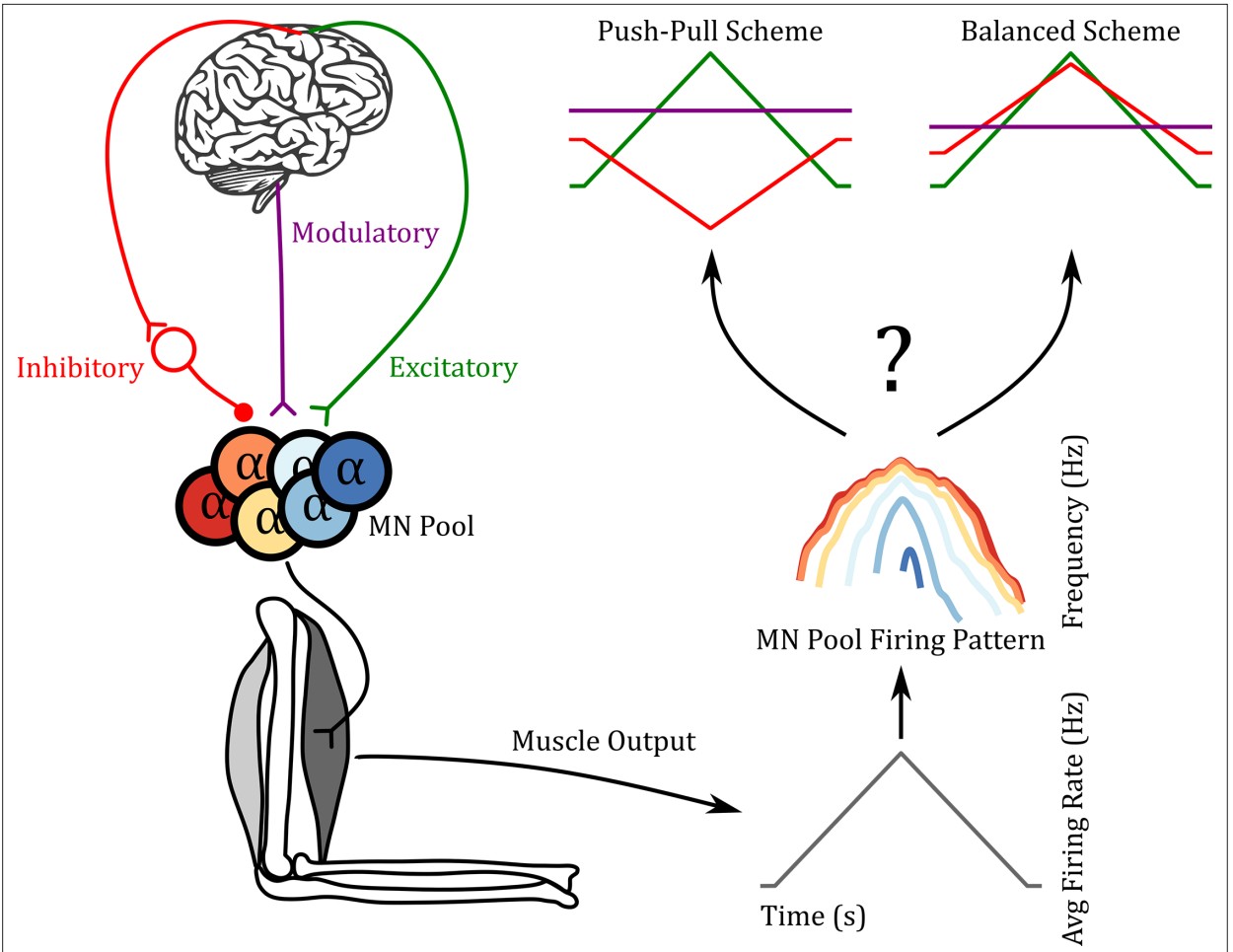

**Figure 2.** Reverse engineering paradigm. The left side shows a schematic of the input pathway to the motoneuron (MN) pool and its subsequent connection to a muscle. The right side shows a schematic of our goal. Can we predict the three inputs of the MN pool given the firing pattern of the MN pool used to generate the muscle output? In the push-pull scheme at the upper right, inhibition (red) decreases as excitation (green) increases. In contrast, inhibition increases with excitation in the balanced scheme. In this work, we show that using the 'MN Pool Firing Pattern' one is able to reverse engineer back to the MN pool inputs.

and amplitudes of the three components of motor commands. Two hypothetical outcomes are illustrated. In both, a triangular form of excitation is applied. The push-pull scheme is a reciprocal organization in which a tonic level of inhibition decreases as excitation increases. The balanced scheme has the opposite organization, inhibition increasing in proportion to excitation. Neuromodulation is assumed to be constant but can vary widely in its level. Although the balanced scheme often is used to drive cortical neurons (*Berg et al., 2007*), it is unknown how these three components interact in motor commands to motoneurons.

The ensemble modeling approach we utilize for RE begins with the estimation of the trajectory of the excitatory motor command. The procedure is illustrated in *Figure 3*. We required all simulations to achieve the same overall output, a triangular waveform for CST (upper left; Methods for the match criterion). For each simulation, we specified the following properties of the input organization:

1. Distribution of excitatory input on to S vs F motoneurons (range of 0.5–2.5; at 0.5 the highest threshold type F motoneuron received half the average input of the lowest threshold type S; at 2.5 the F received 2.5 times the input of the S).
2. Level of neuromodulation: range 0.8–1.2, specified in terms of the maximum conductance of the PIC.

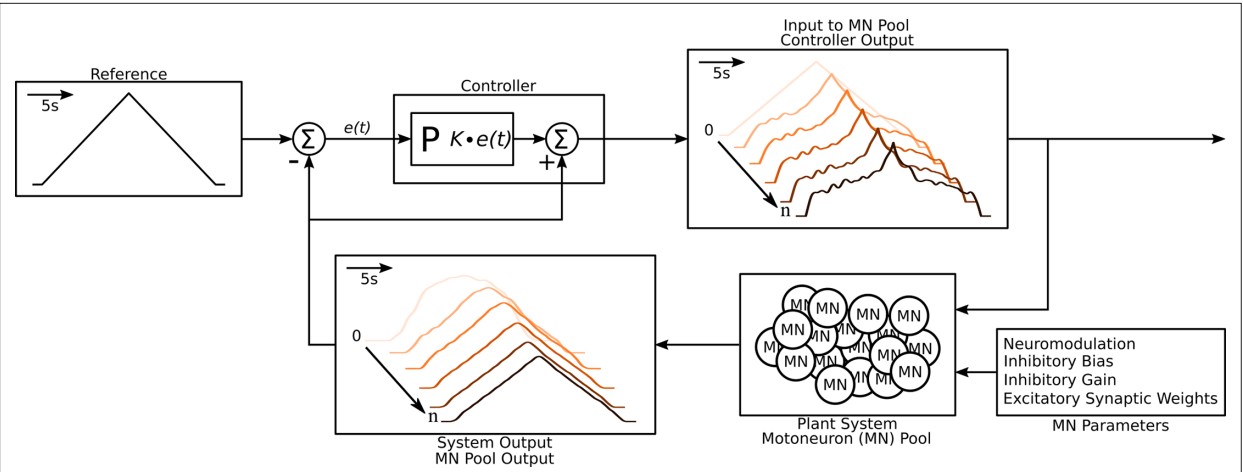

**Figure 3.** Schematic of the algorithm used to approximate the excitatory input needed to produce the triangular output from the motoneuron pool. The schematic resembles a classic controller schematic. The algorithm starts at the 'Input to motoneuron (MN) Pool' box where the light pink triangular input is used as the initial guess to the 'Plant System' box. Using this initial input guess, the MN pool model calculates it first output, seen in the 'System Output' box (light pink trace). This initial output is compared to the 'Reference' Box and the resulting error $e(t)$ is fed into the 'Controller' box. Based on this error the controller calculates a new guess for an input and the cycle is repeated until a satisfactory error is achieved (MSE <1 Hz2). The iterations are shown within the 'Input to MN Pool' and 'System Output' boxes along a left-to-right diagonal scheme to show the evolution of the traces produced by the algorithm. From iteration 0 to iteration $n$ the input to the MN pool morphs to a highly non-linear form to produce a linear output. Finally, this algorithm is repeated over a range of neuromodulation, inhibitory bias, inhibitory gain, and excitatory synaptic weights as shown in the 'MN parameters' box. Taking all the iterations and combinations, the model was run at a total of 6,300,000 times. See subsection Simulation protocol for details.

3. Baseline level of inhibition: This was set to the lowest level that prevented the PIC from generating self-sustained firing (**Figure 1B** shows an example of motoneurons with strong self-sustained firing due to lack of sufficient inhibitory baseline).
4. Pattern of inhibition relative to excitation: This pattern was set to be linearly proportional to excitation via multipliers ranging from +0.7 to –0.7. **Figure 1C1** and **Figure 1C2** illustrate firing patterns resulting from maximum proportional inhibition (multiplier of +0.7) to maximal push-pull (–0.7). At 0.0, the baseline of inhibition remained constant.
5. A physiological level of synaptic noise (see Methods). The RMS value for this noise was the same in all simulations, but was regenerated 30 times to allow estimates of variance for each combination of the above four input properties.

Then, for each simulation, we set the input and noise properties as above and then applied a triangular pattern of excitation to the motoneuron models, starting with a purely triangular form and modifying it in an iterative process until the CST of the motoneurons reproduced the linear target CST (**Figure 3**, flow diagram). We repeated these simulations with different sets of input parameters 315,000 times to form the ensemble model of the motor pool (see section Data preparation for parameter combination details). We used the resulting large data set for successful matches to the target CST to assess the feasibility of RE for both the CST and individual motoneuron firing patterns.

## Solution space for overall output

We hypothesized that the CST, which contains information only about the overall output trajectory, would have a large solution space. It was nonetheless surprising that our process for modifying the shape of the excitatory input command generated good matches to the target CST for every motor command combination we tried (all 315,000 of them (see section Data preparation)). However, as illustrated in by the progression of input forms in **Figure 2** and further elaborated in **Figure 4**, the potent amplification and large nonlinearities in firing patterns induced by PICs usually resulted in highly nonlinear excitatory commands being required to generate the linear CST. The excitatory conductance input that leads to a good match to the target firing rate typically shows a rapid increase to get recruitment started, followed by a drop off during the acceleration phase due to wide-spread PIC activation across the active motor units, followed by a more rapid increase to keep the average

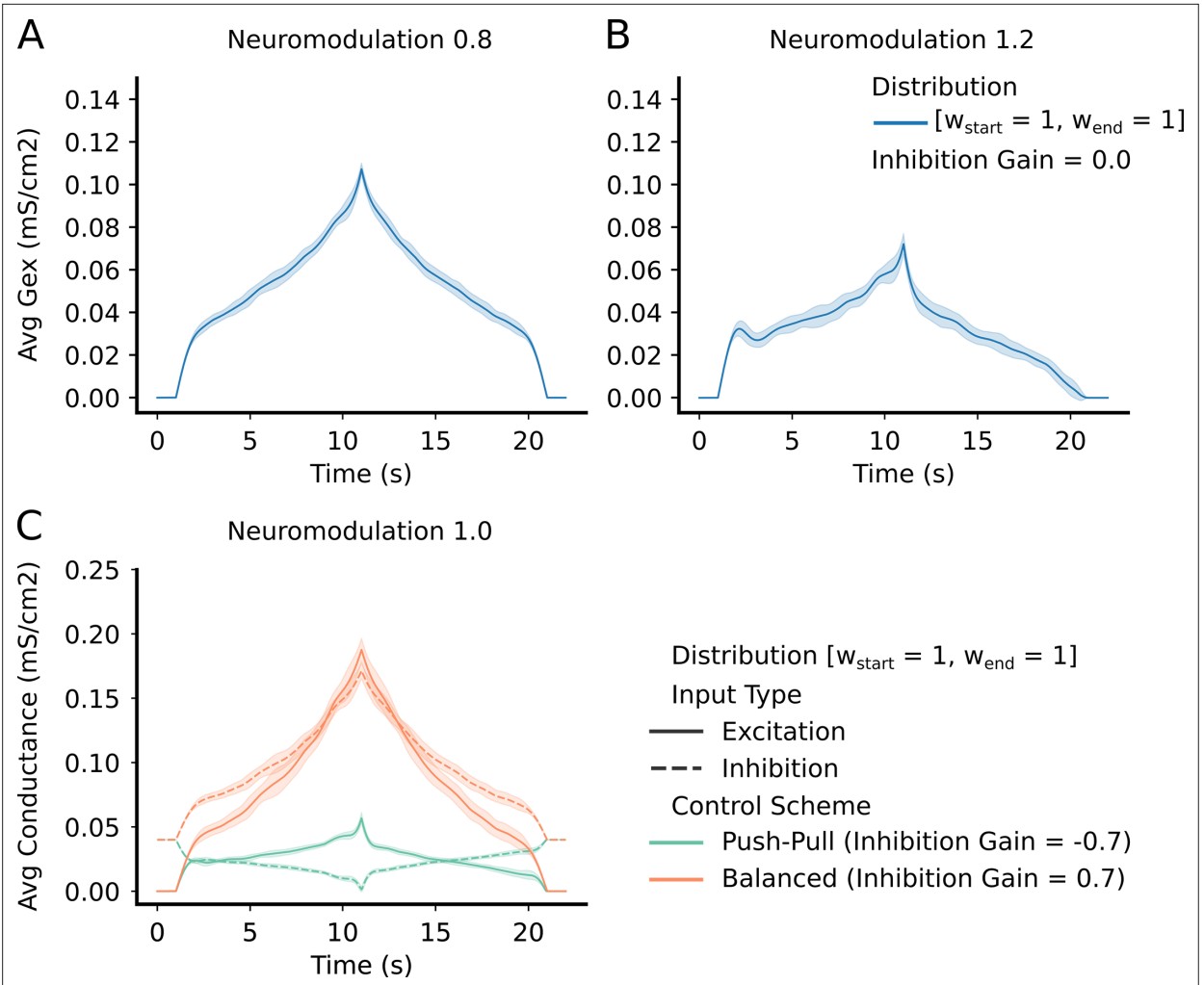

**Figure 4.** *Top Row* – shows the average and std excitatory input to the motoneuron (MN) pool to produce the reference command (*Figure 3* – reference input). (**A**) has a neuromodulation value of 0.8. (**B**) has a neuromodulation value of 1.2. For both (**A**) and (**B**), the inhibitory gain is set to zero and the excitatory weight distribution is equal across the pool (111 configuration - see section Excitatory distribution or MN weights). In short, the lower the neuromodulation the more linear the input command becomes however, the higher the excitatory values becomes. *Bottom Row* - (**C**) - shows the average and std excitation (solid) and inhibition (dotted) for the two extremes of inhibitory schemes (Push-pull in green and Balanced in orange). Neuromodulation is set to 1.0 (middle of range tried) and the excitatory weight distribution is equal across the pool (111 configuration - see section Excitatory distribution or MN weights). The "Push-Pull" inhibitory scheme requires less overall drive to achieve the same output.

firing rate rising in the face of PIC rate saturation. The falling phase of the command was roughly the mirror image of the rising phase. This general trend is evident in all the examples in *Figure 4*.

As expected, as the level of neuromodulation increased, these nonlinear trends increased, but less excitation is required to achieve the same global output. The examples in the upper half of *Figure 4* assumed inhibition was constant. The lower left example in *Figure 4* shows examples of the effects of different organizations of inhibition. Much higher levels of excitation are needed to counteract inhibition when it varies in proportion to the excitatory command (balanced inhibition, red traces), then when it varies inversely (push-pull inhibition, green traces). To quantify the effects of neuromodulation and inhibition on the overall amplitude of excitation, we calculated the area under the curve for the excitation patterns as neuromodulation and inhibition varied. *Figure 5* shows that the amount of excitation required to generate the standard CST target decreased as neuromodulation increased or as inhibition progressively changed from proportional to push-pull. Thus, although the solution space for reproducing the target CST is very wide, encompassing the full range of neuromodulation and inhibition investigated, achieving this wide range required a large changes in the pattern and amplitude of the excitatory drive.

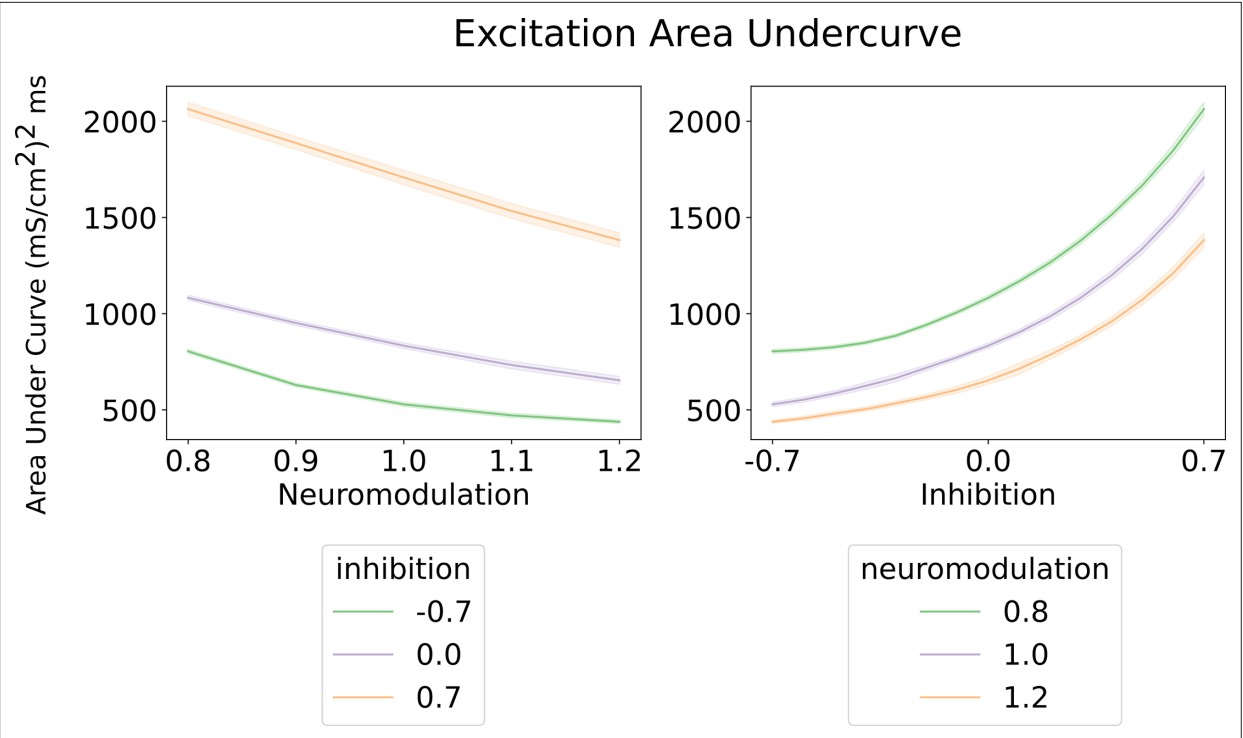

**Figure 5.** Area under the curve for the excitatory inputs calculated by the algorithm (*Figure 3* and Methods). *Left* - Shows the area under the curve of the excitatory input with respect to neuromodulation and inhibitory levels (color scheme). Their inhibitory levels are clearly segregated across the neuromodulation range. The most efficient inhibitory scheme is 'Push-pull' (–0.7) and the least is 'Balanced' (0.7). *Right* - shows the area under the curve of the excitatory input with respect to inhibition for three neuromodulation level. The curves for each neuromodulation level do not intersect across the range of inhibition. The overall trend resembles the results on the left where the most efficent scheme is for high neuromodulation (1.2) and a 'Push-pull' inhibition (–0.7). These curves were calculated using an equally weighted excitatory input.

## The solution space for firing patterns of motor units

If each of the patterns of excitation, neuromodulation, and inhibition that generated the the standard triangular CST generated highly similar motoneuron firing patterns, then the solution spaces for each set of firing patterns would each be large. This firing pattern similarity, however, did not occur. The heat map in *Figure 6* provides a quantitative analysis of one of the main differences in motor unit firing patterns, onset-offset hysteresis this hysteresis was quantified by the $\Delta F$ method developed for studies of real motor unit firing patterns in humans (*Gorassini et al., 2002*) (see Methods and *Figure 7*). The differences in neuromodulation and inhibition produced a huge range of hysteretic firing behaviors, ranging from < 1.0 at low neuromodulation levels coupled to strong proportional inhibition (lower left in *Figure 6*) to > 7.0 for high neuromodulation and strong reciprocal inhibition. *Figure 6* also shows the simulated firing patterns generated by a subset of these combinations (arrows link these patterns to the hysteresis values). As neuromodulation increases, PIC effects (acceleration, attenuation, hysteresis) increase, with push-pull inhibition emphasizing these effects and balanced inhibition suppressing them. These differences for each combination of neuromodulation and inhibition shown in *Figure 6* demonstrate that the solution space for matching a particular set of motor unit firing is much more restricted than for matching the overall CST (see *Figure 4* and *Figure 5*).

Note, however, there is some degree of trade-off for the effects of inhibition and neuromodulation on hysteresis along the diagonals in *Figure 6*. Fortunately, hysteresis is not the only index of motoneuron firing behaviors. *Figure 7* illustrates additional features of motoneuron firing patterns that are easily measured and likely to reflect other effects of excitation, inhibition, and neuromodulation. These features include firing rate acceleration and attenuation (which are also driven by actions of neuromodulation and inhibition on PICs – see *Figure 1C1* and *Figure 1C2*) as well as recruitment and de-recruitment behaviors (sensitive to the distribution of excitation and S vs F motoneurons – see *Figure 1A2*). Each of these other characteristics had patterns of variation with respect to inhibition

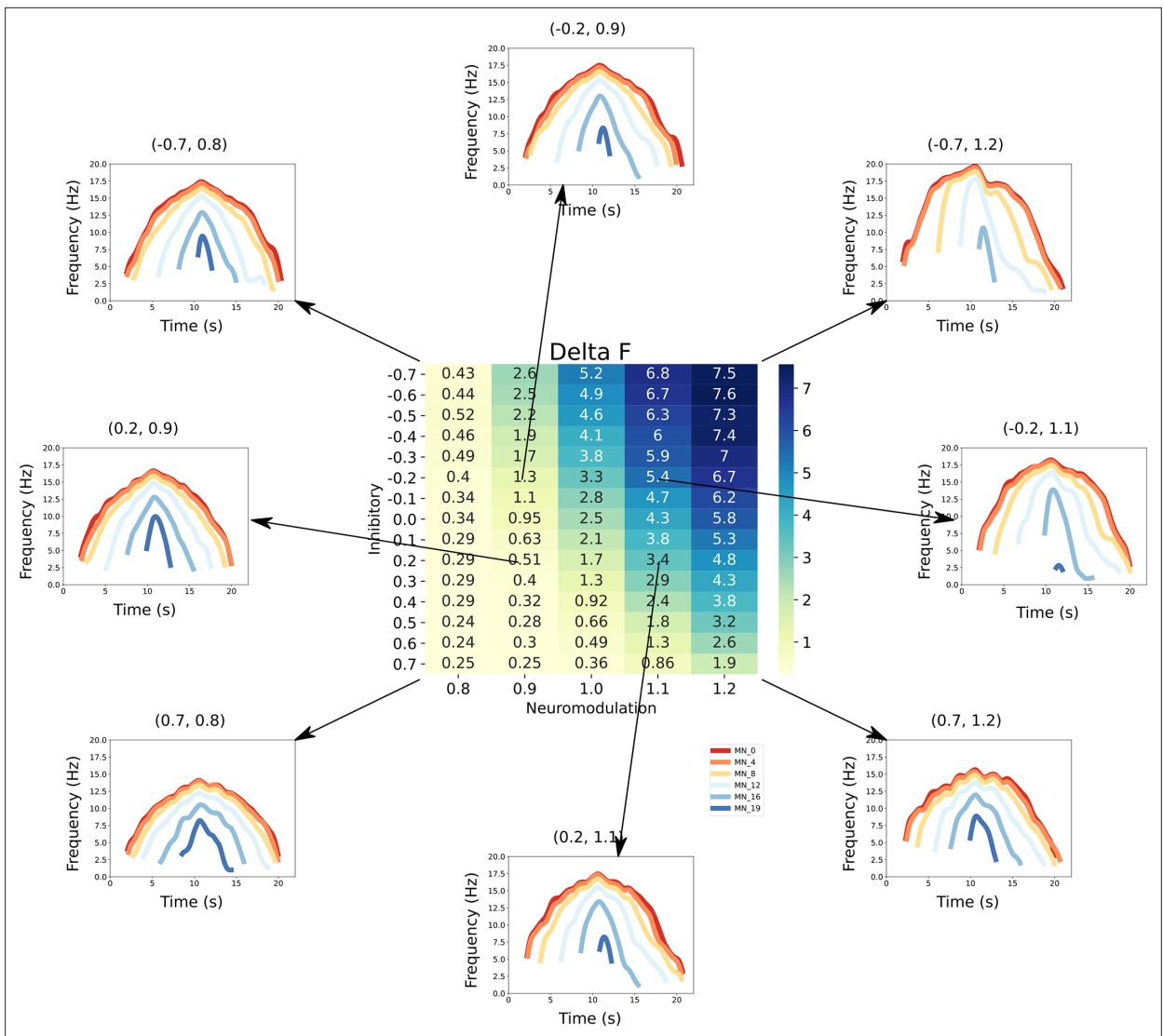

**Figure 6.** This figure links the firing rate shapes of the motoneuron (MN) pool model to the inhibitory and neuromodulatory search space. Heat Map - The central figure is a heat map of the mean $\Delta F$ values of the MN pool model calculated using the firing rate patterns of the motoneurons in the MN pool model (see section Feature extraction). These $\Delta F$ values are organized along the neuromodulation (x-axis) and inhibitory values (y-axis) used in the simulations. Firing Rate Plots - Emanating from the central figure are a subset of the firing rate plots of the MN pool model given a specific Inhibitory and Neuromodulation set (e.g. (0.2, 0.9)). These plots are linked to a $\Delta F$ value with an arrow. For clarity, only a subset of motoneuron firing rates are shown. From visual inspection, the shape of the firing rates differ given the Inhibition and Neuromodulation pair, from more non-linear (e.g. (–0.7, 1.2)) to more linear (e.g. (0.2, 0.9)).

and neuromodulation that varied in comparison to each other and to hysteresis. Therefore, to quantify the effectiveness of RE of motor unit firing patterns to predict the organization of synaptic inputs, we next used multiple regression methods to determine how well combinations of these characteristics of simulated firing properties predicted the level of neuromodulation, the patterns of inhibition and the distribution of excitation to low verse high threshold motoneurons.

## Regression using motoneuron outputs to predict input organization

We utilized several different types of linear regression techniques (see Methods) and one nonlinear technique (Support Vector Regression using a radial basis function kernel (RBF)). In each case, the regression parameters were calculated from a subset of the data (~70% of the total) and then used to predict the remaining subset (~30%). For neuromodulation, all regression methods were similar in their effectiveness.

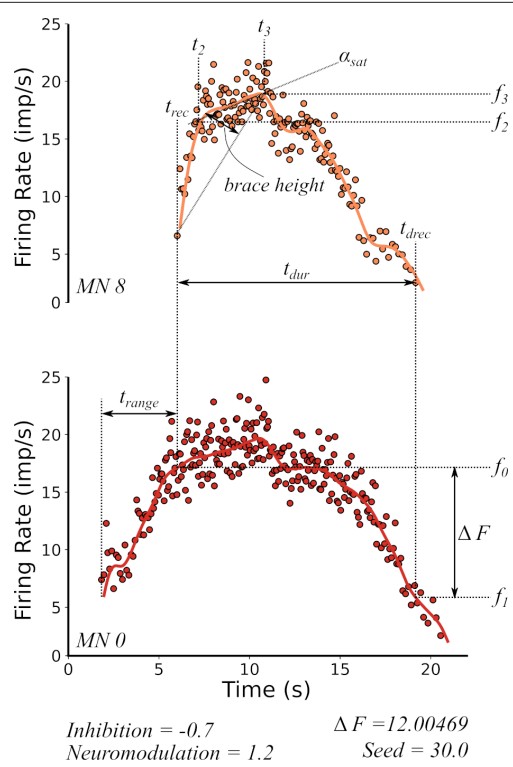

**Figure 7.** Features extracted from the firing rate patterns of each of the motoneurons in the pool. We extracted the following seven features: Recruitment Time ($t_{rec}$), De-Recruitment Time ($t_{drec}$), Activation Duration ($t_{dur}$), Recruitment Range ($t_{range}$), Firing Rate Saturation ($\alpha_{sat}$), Firing Rate Hysteresis ($\Delta F$), and brace–height (*brace–height*). These features are then used to estimate the neuromodulation, inhibition, and excitatory weights of the motor pool (see section Machine learning inference of motor pool characteristics and Regression using motoneuron outputs to predict input organization). See the subsection Feature extraction for details on how to extract features.

*Figure 8A* shows the results for standard linear regression (the dashed red line has slope = 1.0 to indicate perfect predictive performance). For each level of neuromodulation (0.8–1.2), the vertical scatter is due to various combinations of the pattern of inhibition (which ranged from 0.7 to –0.7, as is in *Figure 6*) and of the distribution of excitation to low vs high threshold motoneurons (which ranged 0.5–2.5). The scatter is considerable, but there is also a tendency for points to cluster near the line. As a result, the regression relation accounts for 76% of the variance (i.e. $R^2$ = 0.76; *Table 1* shows the $R^2$ values for all regression methods for the prediction of neuromodulation, inhibition and the distribution of excitation). Thus, prediction of the level of neuromodulation is good and indicates that this regression approach can identify the level of neuromodulation with a resolution of about 0.1 units of neuromodulation within the range 0.8–1.2.

For inhibition, the nonlinear regression was most effective, giving the results shown in *Figure 8B*. Variance accounted for is 0.85, implying a prediction resolution of about 0.23 units of a range of 1.4 units of inhibition. Finally, *Figure 8C* shows the regression results for prediction of the relative weighting ratio for the motoneuron pool, implying that characteristics of motor unit firing patterns can predict input structure with variances accounted for ranging from 76–91%.

As a final step, we analyzed which motor unit firing characteristics had the most predictive power in identifying each of the three input organizations. *Figure 9* provides two perspectives on this question. The left column shows the results of step-wise regression, starting with the characteristic with the strongest predictive power and then progressively adding the others. For neuromodulation, all regression methods behaved similarly, in that the first two characteristics (brace-height and $\Delta F$ see *Figure 7*) was highly effective in reducing error in the regression prediction, with little addition benefit from adding the others. For inhibition prediction via the linear regression models, $\Delta F$ was most effective, followed by $\alpha_{sat}$, with all others being ineffective. This is shown in the mutual information ranking (right side) in *Figure 9*, which ranks the characteristics (features) by the contribution of new information to the regression model. The left side of *Figure 9* ranks the features by the F-statistic, which does not necessarily imply that new information is introduced into the regression model. This is shown by the very subtle improvements in the error (mean squared error) as features are cumulatively added to the regression model, until $t_{drec}$ and $\alpha_{sat}$ are added to the model.

However, the nonlinear regression method (RBF SVR) was more effective than any of the linear methods. For this method, each of the firing characteristics had a notable effect in reducing error. For linear regression for prediction of excitation, $t_{drec}$ (the average time when units were de-recruited) was the most impactful, and adding the other characteristics brought little improvement. The right column in *Figure 9* shows mutual information for each characteristic, which reflects how well each one predicts the input behaviors on its own. As expected, the order of effectiveness of prediction in these plots is identical to that for the step-wise regression.

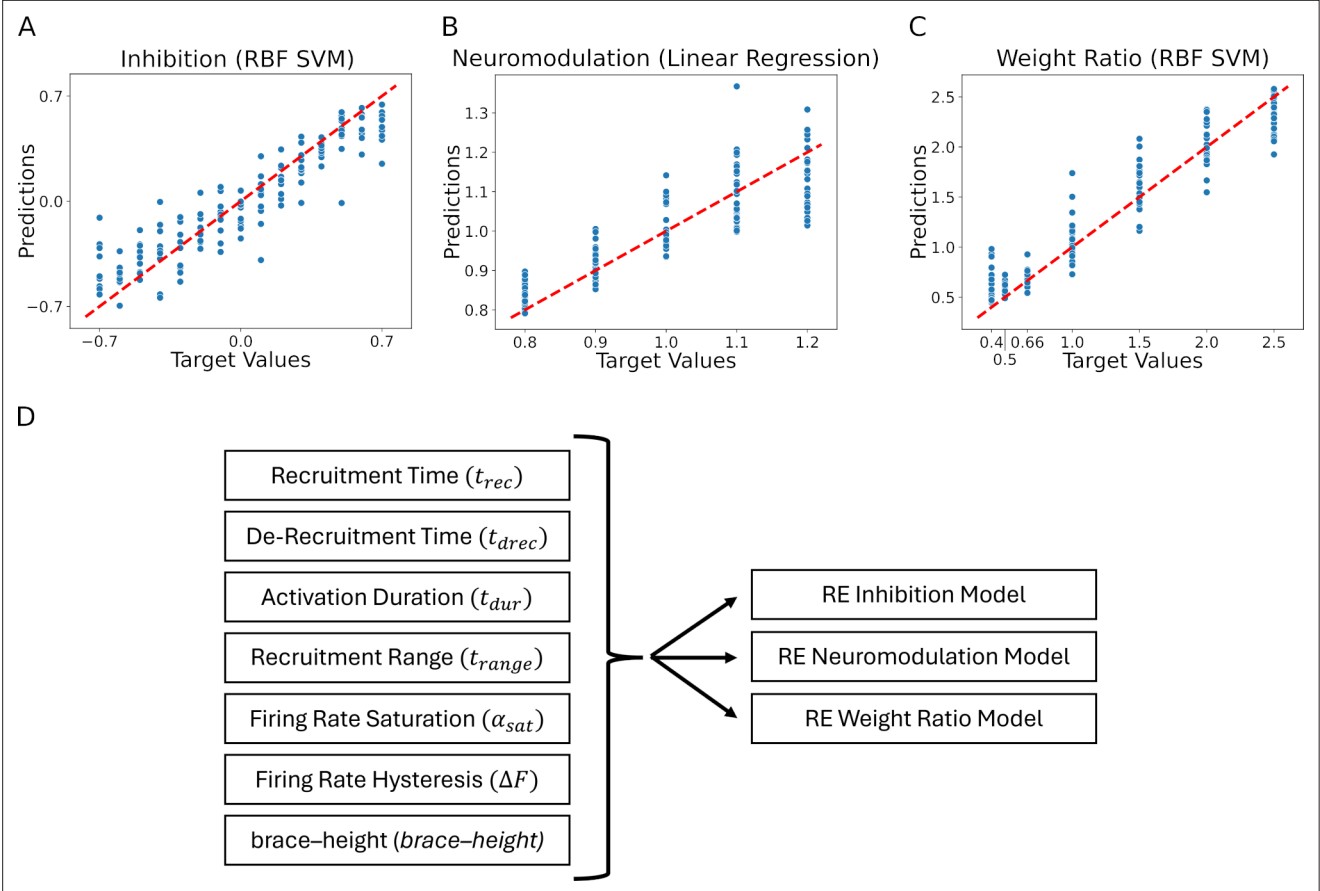

**Figure 8.** Residual plots showing the goodness of fit of the different predicted values: (**A**) Inhibition, (**B**) Neuromodulation, and (**C**) excitatory Weight Ratio. The summary plots are for the models showing the highest $R^2$ results in **Table 1**. The predicted values are calculated using the features extracted from the firing rates (see **Figure 7**, section Machine learning inference of motor pool characteristics and Regression using motoneuron outputs to predict input organization). Diagram (**D**) shows the multidimensionality of the reverse engineering (RE) models (see Model fits) which have seven feature inputs (see Feature Extraction) predicting three outputs (Inhibition, Neuromodulation, and Weight Ratio).

Overall, neuromodulation is best predicted by brace-height and $\Delta F$. Inhibition is also predicted by $\Delta F$ while *brace–height* is ineffective. Excitation (excitatory weight ratio) is best predicted by parameters that assess recruitment and derecruitment timings and are less informed in prediction by $\Delta F$ and *brace–height*. These different roles for RE prediction of input make sense in terms of the physiological effects of the three types of inputs on motoneuron firing patterns (see Discussion).

## Discussion

Our simulation shows that the solution space for a motor output variable that reflects the net action of all active motoneurons is indeed very broad. The variable investigated here, the CST, has been shown to be closely proportional to muscle EMG and muscle force (**Thompson et al., 2018**), which are widely assessed in experimental paradigms in both animals and humans. Our simulations, however, indicate that RE analyses of CSTs, EMGs, and forces are likely to fail due to the problem of non-uniqueness. In contrast, motoneuron firing patterns provide a much greater level of information about motoneuron inputs. Our simulations showed that the solution space for a set of motoneuron firing patterns is reasonably small, allowing predictions that account for 75–90% of the variance in the level of neuromodulation, the pattern of inhibition, and the distribution of excitation to S vs F motoneurons.

### Generalization

Array electrodes are extensively used in many parts of the CNS to record firing patterns of populations of neurons in awake behaving animals (see Introduction). Moreover, highly realistic models of neurons

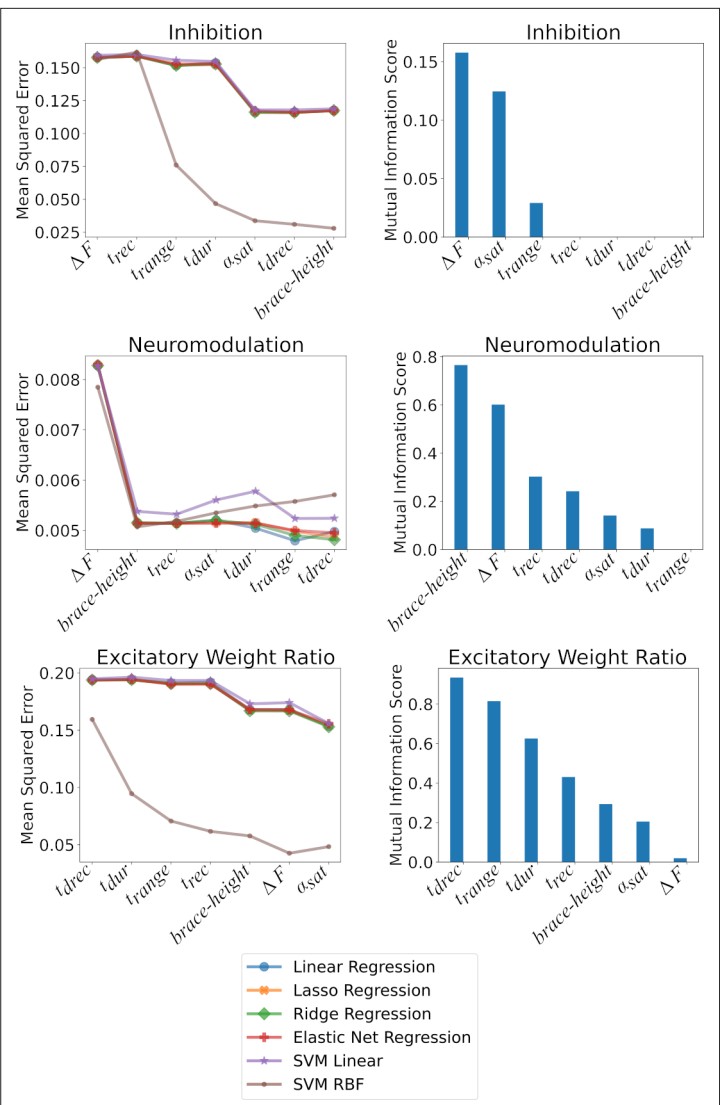

**Figure 9.** Shows the impact of the features on the models at predicting Inhibition, Neuromodulation, and the excitatory Weight Ratios. The plots are split along two columns showing two types of complementary analysis: F-Statistic (left) and Mutual Information (right). *Left* - The left column shows the impact of the features on the MSE as ranked by their $F$ value. The effect of the features are cumulative along the x-axis such that the value of the MSE at a given position in the x-axis is calculated using the features up to that point. Each of the colored traces represents one of the models used to predict the three outcomes and follows the trends found in ***Table 1***. *Right* - The right column shows the mutual information of each feature with respect to the outcome: Inhibition, Neuromodulation, and excitatory Weight Ratio (Top to Bottom). The features are ranked from best to worst along the x-axis. The ranking matches the MSE ranking in the right column. See the section Ranking features to see details on how these techniques were implemented.

and their interconnections in many systems are already available and undergoing rapid development, including supercomputer implementation (***Einevoll et al., 2019***). Since models and firing patterns are the fundamental elements for the success of RE in the simulations presented here, this same type of approach may be effective in many other neural systems. In each neural system, however, the success of RE of firing patterns into input structures as simulated here is likely to depending on the particular interactions between neuromodulation, inhibition, and excitation. For motoneurons the nonlinearities induced by PICs in motoneuron firing patterns have proven to be a major advantage for RE. These nonlinearities impart distinctive characteristics to firing patterns (acceleration, attenuation, hysteresis – see ***Figure 1C1***, the firing pattern at the upper right in ***Figure 6***), which are easy to quantify (see

**Table 1.** $R^2$ values.

Rows are the predicted values. Columns are the models. The $R^2$ values in bold represent the best model fit: Inhibition is best predicted using the radial basis function kernel (RBF) SVM model, Neuromodulation is best predicted by the Ridge model, and Excitation Weight Ratio is best predicted by the RBF SVM model.

| Name | Linear regression | Lasso | Ridge | Elastic net | Linear SVM | RBF SVM |
|---|---|---|---|---|---|---|
| Inhibition | 0.36 | 0.36 | 0.36 | 0.36 | 0.35 | 0.85 |
| Neuromodulation | 0.75 | 0.76 | 0.76 | 0.75 | 0.74 | 0.71 |
| Excitation Weight Ratio | 0.72 | 0.72 | 0.72 | 0.72 | 0.72 | 0.91 |
| Average | 0.61 | 0.61 | 0.61 | 0.61 | 0.60 | 0.83 |

*Figure 7*). Moreover, because the PIC is highly sensitive to inhibition, these same firing characteristics provided effective indices of the temporal pattern of inhibition. Furthermore, because these relations between neuromodulation and inhibition were different for each firing characteristic (see Supporting Information Figure S1), multiple regression using these characteristics was effective. It seems likely that if motoneurons were nearly linear input-output processors, non-uniqueness would be a much greater problem for RE.

## Push-pull vs balance motor command

As we defined earlier (see section Ensemble modeling for reverse engineering) our model explores range of possible combinations of excitatory and inhibitory motor commands from push-pull to balanced, for a motoneuron pool to produce a triangular ramp output (*Figure 2*). The push-pull scheme is a reciprocal organization in which a tonic level of inhibition decreases as excitation increases. The balanced scheme has the opposite organization, with inhibition increasing in proportion to excitation (see *Powers and Heckman, 2017*, *Johnson et al., 2017* for full reviews detailing both types of schemes). The question remains, which of these schemes is being used in normal human motor behavior? Based on our results we would speculate that the most likely scheme would be push-pull, as the upper right motoneuron pool discharge pattern shown in *Figure 6* seems the most realistic when compared to human data from the lower limb (see for example *Beauchamp et al., 2022*, *Afsharipour et al., 2020*). We have, however, emphasized that different muscles might require different types of motor command schemes (*Johnson et al., 2017*), and herein lies the power of this RE technique as the experimenter can use the RE approach detailed here to identify the scheme that best fits their human data.

## Muscle EMG vs single motor unit firing patterns

Muscle EMG is the electrical signal generated by the summed action potentials of the muscle fibers of all active motor units. EMG is easy to measure in many types of motor behaviors in humans and other animals and provides an effective measurement of the timing and overall pattern of the neural output from the spinal cord to muscle. However, the simulations here and in our previous work (*Powers et al., 2012*; *Powers and Heckman, 2017*) show that EMG does not provide information about the organization of motor commands that produce spinal motor output. On the other hand, because motoneurons action potentials are one-to-one with those of their muscle fibers in healthy humans and other animals, motor unit firing patterns are equivalent to motoneuron firing patterns. While motor unit firing patterns are more technically difficult to measure than whole muscle EMG, the accessibility of muscle makes these measurements highly feasible. In fact, motor unit firing patterns in humans were recorded at the very onset of single-neuron studies, over 90 years ago (*Adrian and Bronk, 1929*). The likelihood that this single neuron information provides deep insights into the structure of motor commands has thus long been appreciated (*Duchateau and Enoka, 2011*). Many such studies were essential for establishing the prevalence of Henneman's size principle of recruitment (reviewed in *Heckman and Enoka, 2012*). Many studies also focused on inferring the time course of EPSPs and IPSPs driving motoneuron responses to transient ionotropic inputs during steady contractions

(reviewed in *Türker and Powers, 2005*). The distinctive effects of PICs on firing have also been a focus, with the $\Delta F$ method developed by Gorassini and colleagues for quantifying PIC-induced hysteresis having become a standard in the field *Gorassini et al., 2002*. However, intracellular current and voltage clamp studies have demonstrated that the PIC is highly sensitive both to neuromodulatory input (*Hounsgaard et al., 1988*; *Lee and Heckman, 2000*; *Lee and Heckman, 1999*) and inhibition (*Hyngstrom et al., 2008*; *Kuo et al., 2003*). Consistent with this, $\Delta F$ does not provide good discrimination of neuromodulation vs inhibition in our simulations (*Figure 9*). In contrast, and consistent with our previous study (*Beauchamp et al., 2023*), the brace–height parameter was sensitive to neuromodulation but insensitive to inhibition, likely because it is normalized for peak firing. In general, our simulations show that no one motor unit firing characteristic is sufficient for estimation of input structure and thus a form of RE is necessary.

## Limitations

Ensemble model approaches for RE are obviously reliant on the biological accuracy of the constituent models. Our models have been carefully tuned to replicate experimental data from years of current and voltage clamp studies of the integration of neuromodulatory, inhibitory, and excitatory inputs motoneurons in feline preparations. We made a few adjustments to these models to more closely replicate the slow firing rates in human motor units but will likely need to further adjust parameters like the spike AHP and the PIC voltage threshold. Our RE was not applied for identification of the amplitude or time course of the excitation. The peak excitatory conductance for each simulation strongly depends on the level of neuromodulation and the pattern of inhibition (*Figure 4*). This correlation was built into our simulations to match the CST for each combination of neuromodulation/inhibition and so excitation was not an independent parameter. Further simulations in which we relax this constraint or allow inhibition and/or neuromodulation to vary for a given pattern of excitation may prove interesting. It will also be interesting to expand our range of neuromodulation. At the high end, this expansion will have to be limited, as controlling such large PICs is difficult. But exploring whether RE works as at lower neuromodulation levels where input-output processing is nearly linear would reveal whether this simpler behavior limits RE effectiveness.

## Summary and future application

Although our models likely need modification before we begin RE of human motor unit firing patterns, they were well situated to the goal of this study, which was to investigate the feasibility of RE of neuron firing patterns despite the strong tendency for neural systems to be non-unique. In addition, it would seem that the success of our RE approach suggests that this technique can be used to gain insight into the motor commands underlying different motor unit firing patterns associated with different motor tasks with or without disease states (e.g. *Mottram et al., 2014*).

# Materials and methods

## Motoneuron pool model

The motoneuron pool consists of 20 model motoneurons with a range of intrinsic properties. This motoneuron pool size was chosen to reflect the typical sample size of motor units discriminated based on surface EMG array recordings. Each model motoneuron consisted of a soma compartment and four dendritic compartments, each coupled to the soma. The passive properties of the compartments, such as, size, capacitance, and passive conductance were derived from works by described by Kim and colleagues (*Kim et al., 2009*; *Kim and Jones, 2012*).

Spike conductances ($g_{Na}$ and $g_K$) and conductances mediating the medium AHP were inserted into the soma compartment and a calcium conductance mediating the slowly-activating PIC was inserted into each of the dendritic compartments. In addition, a hyperpolarization-activated mixed-cation (HCN) conductance was inserted into all compartments. Conductance densities, kinetics and steady-state activation curves were originally tuned to recreate the range of input-output behavior recorded in medial gastrocnemius (MG) motoneurons in decerebrate cats, as described in *Powers and Heckman, 2017*.

We made four modifications to model parameters for the cat MG pool in order to produce the lower firing rates and less than full recruitment typically observed during moderate voluntary contractions in

human subjects. First, the range of values governing excitability were restricted to the first 75% of the original recruitment range (i.e. the parameters of motoneuron 20 in the new model corresponded to motoneuron 15 in the original parameter set). Second, AHP durations and amplitudes were increased by increasing the values of the time constant of calcium removal for the calcium-activated potassium conductance (from 60–10 ms in the original model to 90–57 ms). The larger and longer AHPs acted to oppose early PIC activation during increasing excitatory synaptic drive, so we hyperpolarized the PIC half-activation threshold (from –40 to –37 mV in the original model to –42 to –40.4 mV). Finally, in the original model the slow decay of PICs observed in high threshold MG motoneurons (*Lee and Heckman, 1996*) was replicated by including a voltage-dependent inactivation process. We found that this process limited the firing rate hysteresis to values below those typically seen in human motor unit recordings, so we eliminated PIC inactivation in the present model pool. All simulations were run using the Python interface to the NEURON simulator (*Hines and Carnevale, 1997*). NEURON files specifying motoneuron pool parameters, conductance mechanisms, and protocols on ModelDB with accession number 2017005.

## Simulation protocol

### Inputs to the motoneuron pool

The target motoneuron pool output, representing the average firing rate across the entire pool of 20 motoneurons was 22 s in length and consisted of 1 s delay followed by a linear rise to a peak value of 16 imp/s over the next 10 s, and a linear decline back to zero in 10 s (*Figure 3*, Reference *Ref*). A scaled version of this command, $I_{ex} = 0.6Ref$, (*Figure 3*, light salmon-colored with the value 0 to its bottom left) was used as the first guess of the time course of the excitatory input needed to produce the target output. For each iteration an inhibitory conductance input ($I_{in}$) was also applied according to the following equation:

$$I_{in} = G_{in}I_{ex} + b_{in} \tag{1}$$

where the excitatory input $I_{ex}$ is multiplied by a gain $G_{in}$ and biased by $b_{in}$ as defined by:

$$b_{in} = 6.25R_{mod}^2 - 1.25R_{mod} - 1 \tag{2}$$

where $R_{mod}$ is the neuromodulation value, set in this work, from 0.8 to 1.2 at increments of 0.1 (i.e. [0.8:0.1:1.2]). The inhibition is, therefore, coupled to the neuromodulation through its bias, $b_{in}$, and was set to value just sufficient to ensure PIC deactivation at the end of the command. The inhibitory gain, $G_{in}$, is varied from –0.7 to 0.7 in increments of 0.1 thus traversing the possible control schemes push-pull inhibition to balanced inhibition. Examples of both control schemes can be seen in *Figure 2* (green for excitatory and red for inhibitory).

Changes in neuromodulation were simulated by changing the density of dendritic PIC channels: from 80–120% of the standard density in 10% increments. Mathematically, the MN model was altered by changing the current $I_{Ca_L}$ of the L-type calcium channel according to these equations:

$$I_{Ca_L} = g_{Ca_L}m_{Ca_L}(V - E_{Ca_L}) \tag{3}$$

$$g_{Ca_L} = R_{mod}\bar{g}_{Ca_L} \tag{4}$$

where $I_{Ca_L}$ is the current for the L-type calcium channel, $g_{Ca_L}$ is the conductance of the L-type calcium channel, $m_{Ca_L}$ is the activation gate, $V$ is the compartment voltage, and $E_{Ca_L}$ is the reversal potential for the L-type calcium channel. $R_{mod}$ is the gain factor that changes the maximum conductance – $\bar{g}_{Ca_L}$ – of the L-type Calcium channel with the following values [0.8, 0.9, 1.0, 1.1, 1.2].

Finally, filtered Gaussian noise was added to both the excitatory and inhibitory commands with a standard deviation that varied in proportion to the square root of the mean level and a decay time constant of 20 ms to reflect the fact that most of the power in the synaptic input to motoneurons is thought to be in the low frequency (<10 Hz) range (*Farina et al., 2014*).

### Feedback and optimization

We used an iterative optimization procedure to find the motoneuron pool input needed to produce the 'Reference' output as shown in *Figure 3*. We opted for a feedback type of optimization method to converge onto the solution.

For the feedback to work, an error term is calculated between the output of the MN pool and the 'Reference' (see the 'Controller' *Figure 3*). The output of the MN pool is the average of the 20 MN pool calculated by convolving each model spike train with a 2 s Hanning window with the unit area, adding up the result across all 20 spike trains, and dividing by 20:

$$I_{ex_n} = \frac{1}{20} \sum_{i}^{20} \left( F(t)_i * H(t) \right) + Ke(t) \tag{5}$$

where $I_{ex_n}$ is the new excitatory input, $n$ is the iteration in the feedback loop, $F(t)_i$ is the firing rate of a MN $i = [0 : 20]$ in the 20 MN pool, $H(t)$ is the 2 s Hanning window, $e(t)$ is the error and $K$ is the proportional gain (usually 20%). The resulting $I_{ex_n}$ (the average firing rate response) plus the error $e(t)$ is then used as the new excitatory command to the MN Pool. This is shown by the series of traces leaving the 'Plant System - Motoneuron (MN) pool' in *Figure 3*. Using $I_{ex_n}$, a new inhibitory command is also calculated using *equation 1* and is subsequently used in the new computational iteration $n$.

This process was repeated until the mean squared error between the output and the target was less than 1 Hz$^2$ (generally less than 0.5 Hz$^2$). In the case shown in *Figure 3*, good convergence was reached by the fifth iteration.

This iterative matching procedure was repeated for 225 different input combinations: five levels of neuromodulation, three different distributions of excitatory input, and fifteen different levels of inhibition.

## Excitatory distribution or MN weights

The density of the excitatory synaptic conductance was either uniform across all cells, weighted toward low threshold units (synaptic weight onto lowest threshold unit 2 X that of highest threshold units), or weighted toward high threshold units (high weight 2 X low). The parameters governing cell excitability were skewed to produce a greater proportion of low threshold units, so synaptic weighting showed a similar skew according to the following equation:

$$w_{MN\#} = w_{start} + \left( \frac{MN\#}{\#MN - 1} \right)^2 (w_{end} - w_{start}) \tag{6}$$

where $w_{MN\#}$ is the weight of the given motoneuron, $\#MN$ is the number of the MN in the pool (e.g. 20), $w_{end}$ and $w_{start}$ are the weight range. For weights favoring low threshold units, $w_{start} = 2.5$ and $w_{end} = 1.0$ and for the opposite $w_{start} = 1.0$ and $w_{end} = 2.5$. In total we used seven different types of weight configurations: $[w_{start}, w_{end}] \iff [1, 1], [1.5, 1], [2, 1], [2.5, 1], [1, 1.5], [1, 2], [2.5, 1]$.

## Noise seed

We chose to calculate the model using 30 different noise seeds. We assumed that each seed would represent a 'unique' individual and allow for the use of averages and variance in our interpretation of the results. *Note*: The noise seed was not reset for each of the motoneurons in the pool. The synaptic noise is thus coupled for each of the motoneurons which fits with the 'common drive' argument (see section Common input structure, differences in motoneuron properties).

## Massively-parallel data generation methods

Since the majority of the simulation is able to operate independently as the simulations for each of the parameter sets (noise seeds, inhibitory gains, and neuromodulation) did not need to feed into each other, we were able to run this simulation in a perfectly parallel way. Each unique parameter set was simulated using one node (computer) in a supercomputing cluster, which produced a dataset file of the simulation results (see Algorithm 1 in Supporting Information).

Each node simulated one motor pool, with the computation of each motoneuron run on each processor on the compute node (see Algorithm 2 in Supporting Information).

To implement this setup, we utilized the MPI interface in NEURON 7.7 *Hines and Carnevale, 1997* and Python 3.6.7 running on the Bebop supercomputer (Intel Broadwell nodes) at Argonne National Laboratory, (*Appendix 2—figure 2*). Each combination tuple tuple_comb contains a particular ginmult, nmmult, ginadd, and random_seed (if noise is used).

## Feature extraction

For each simulation input combination, we analyzed the individual motoneuron discharge patterns to extract seven features: Recruitment Time ($t_{rec}$), De-Recruitment Time ($t_{drec}$), Activation Duration ($t_{dur}$), Recruitment Range ($t_{range}$), Firing Rate Saturation ($\alpha_{sat}$), Firing Rate Hysteresis ($\Delta F$), and brace–height (*brace–height*). The calculation of these quantitative measures is illustrated in *Figure 7*, which shows the instantaneous discharge rates (dots) of a higher (upper panel) and lower threshold (lower threshold) motoneuron, along with the smoothed firing rates calculated by convolving the spike times with a 2 s Hanning window (continuous lines).

*Recruitment Time* ($t_{rec}$): Recruitment time is the time of the first instantaneous discharge rate of a given motoneuron.

*De-Recruitment Time* ($t_{drec}$): De-Recruitment time is the time of the last instantaneous discharge of a given motoneuron.

*Activation Duration* ($t_{dur}$): Activation duration is the difference between de-recruitment time and recruitment time:

$$t_{dur} = t_{drec} - t_{rec} \tag{7}$$

*Recruitment Range* ($t_{range}$): The recruitment range is the difference between the recruitment time of the last and the first motoneuron in the motor pool.

*Firing Rate Saturation* ($\alpha_{sat}$): The motoneuron firing rate is characterized by an initial acceleration phase that lasts about 1 s followed by partial saturation of firing rate. The degree of this saturation is quantified by calculating the rate of firing rate increase from 1 s after recruitment to the target peak (11 s):

$$\alpha_{sat} = \frac{f_3 - f_2}{t_3 - t_2} \tag{8}$$

where $f_3$ is the firing rate at 11 s, $f_2$ is the firing rate 1 s post recruitment ($t_{rec}$), $t_3$ is a time at target peak 11 s and $t_2$ is 1 s post recruitment.

*Firing Rate Hysteresis* ($\Delta F$): A motoneuron rate hysteresis can be quantified using the $\Delta F$ measure introduced by Gorassini and colleagues (*Gorassini et al., 2002*). $\Delta F$ is defined as the difference in the smoothed firing rate of a lower threshold reporter motoneuron (*Figure 7* - bottom) at the onset ($t_{rec}$, $f_0$) and offset ($t_{drec}$, $f_1$) of the higher threshold motoneuron (*Figure 7* - top):

$$\Delta F = f_0 - f_1 \tag{9}$$

brace–height (*brace − height*): Brace height is a measure of the non-linear portion of the rising phase of the firing rate of the motoneuron (see *Figure 7* - top trace). It is the maximum perpendicular distance from the line linking the firing rates at recruitment time ($t_{rec}$) to $t_3$, the time at which the triangular 'Reference' reaches its maximum (11 s). This distance is calculate by solving the geometric problem of finding the 'Distance from a point to a line'. This analysis is further detailed by *Beauchamp et al., 2023*.

We restricted the calculation to motoneuron pairs for which the recruitment time difference between the two was more than 1 s. This restriction was imposed to ensure that PIC activation in the reporter unit had reached near steady state, so that further changes in the reporter discharge rate primarily reflected changes in synaptic drive (cf. *Powers et al., 2008*).

## Machine learning inference of motor pool characteristics

### Data preparation

We ran the model for a range of neuromodulation (5), inhibition (15), excitatory weights (7), and noise seeds (30) for a total of 15,750 combinations. Given 20 motoneuron per pool and 20 optimization iterations see *Figure 3* or section Feedback and optimization we ran a grand total of 6,300,000 simulations.

Post simulations, we extracted the features from the firing rates of each motoneuron see section Feature extraction and organized them according to the neuromodulation, inhibition, excitatory weight, and noise seed. We then first calculated the average of these features accordingly and then took an average of these averages with respect to the noise seed. In final, the was an average of

the features grouped by neuromodulation, inhibition, and excitatory weight. *Appendix 2—figure 1*. shows the average of each feature in a heat map configuration for the excitatory weight distribution $[w_{start}, w_{end}] = [1, 1]$. This data was the used subsequently (see section Model fits and Ranking features).

## Model fits

One of the goals of our RE approach is to infer the characteristics of the input to the motoneuron pool based on our measures of motor unit output. The numeric features extracted from the discharge patterns are used to create five-dimensional input data for the models. The dimensions were the mean $\Delta F$ value, times of recruitment ($t_{rec}$) and derecruitment ($t_{drec}$), the firing rate saturation degree ($\alpha_{sat}$), the recruitment range ($t_{range}$), and brace–height ($brace - height$). The target values to predict were the simulation input values, namely level of neuromodulation, inhibition, and excitatory weight. Since all target values to be predicted in this study were continuous numeric values, the RE process was posed as a regression task.

Since the dataset is relatively small (<1000 instances) and has only 5 dimensions, linear methods and kernel methods were chosen for predictive analysis. More complex models, such as tree-based methods and neural networks, were omitted due to inadequate training data instances and interpretability.

Linear regression models using standard least-squares, Lasso, Ridge, and ElasticNet methods, and support vector models using linear and radial basis function kernels were created for each simulation input value to be predicted using the mean squared error as the minimization cost function.

## Ranking features

To measure the performance of linear regression against the firing features (see section Feature extraction), we use a greedy approach towards incorporating more and more features into the linear regression model. For this approach, the F-statistic (linear regression test) is used as the scoring function to order-rank the features for incorporation into a linear regression model.

We used mutual information (MI) (*Ross, 2014*; *Kraskov et al., 2004*; *Kozachenko and Leonenko, 1987*) to rank the dependence of the firing rate features (see section Feature extraction) to the inhibition, neuromodulation, and excitatory weight ratio levels of the motoneurons in the pool model.

# Acknowledgements

This research used resources of the Argonne Leadership Computing Facility, which is a DOE Office of Science User Facility supported under Contract DE-AC02-06CH11357. We gratefully acknowledge the computing resources provided on Bebop, a high-performance computing cluster operated by the Laboratory Computing Resource Center at Argonne National Laboratory. This research was supported by the National Institute of Neurological Disorders and Stroke of the National Institutes of Health under awards R01NS062200 and R01NS125863 as well as the National Science Foundation NSF DBI 2015317 as part of the NSF/CIHR/DFG/FRQ/UKRIMRC Next Generation Networks for Neuroscience Program. We also gratefully acknowledge Rochelle O Bright for proofreading this work.

# Additional information

## Funding

| Funder | Grant reference number | Author |
| --- | --- | --- |
| National Institute of Neurological Disorders and Stroke | R01NS062200 | Matthieu K Chardon<br>J Andrew Beauchamp<br>Randall K Powers<br>Charles J Heckman |
| National Institute of Neurological Disorders and Stroke | R01NS125863 | Matthieu K Chardon<br>J Andrew Beauchamp<br>Randall K Powers<br>Charles J Heckman |

| Funder | Grant reference number | Author |
| --- | --- | --- |
| DOE Office of Science User Facility | DE-AC02-06CH11357 | Marta Garcia |
| National Science Foundation Next Generation Networks | NSF DBI 2015317 | Matthieu K Chardon Charles J Heckman |

The funders had no role in study design, data collection and interpretation, or the decision to submit the work for publication.

### Author contributions
Matthieu K Chardon, Y Curtis Wang, Conceptualization, Resources, Data curation, Software, Formal analysis, Supervision, Validation, Investigation, Visualization, Methodology, Writing – original draft, Project administration, Writing – review and editing; Marta Garcia, Resources, Data curation, Software, Writing – review and editing; Emre Besler, Software, Validation, Writing – review and editing; J Andrew Beauchamp, Methodology; Michael D'Mello, Software; Randall K Powers, Conceptualization, Software, Formal analysis, Supervision, Methodology, Project administration, Writing – review and editing; Charles J Heckman, Conceptualization, Supervision, Funding acquisition, Methodology, Writing – original draft, Project administration, Writing – review and editing

### Author ORCIDs
Matthieu K Chardon ⬡ https://orcid.org/0000-0001-5723-3217
Charles J Heckman ⬡ https://orcid.org/0000-0002-1324-7278

Reviewer #1 (Public Review): https://doi.org/10.7554/eLife.90624.3.sa1
Author response https://doi.org/10.7554/eLife.90624.3.sa2

## Additional files

### Supplementary files
• MDAR checklist

### Data availability
The current manuscript is a computational study, so no data have been generated for this manuscript. Modelling code can be found at https://modeldb.science/2017005.

The following dataset was generated:

| Author(s) | Year | Dataset title | Dataset URL | Database and Identifier |
| --- | --- | --- | --- | --- |
| Chardon et al | 2023 | Reverse Engineering MN Pool | https://modeldb.science/2017005 | ModelDB, 2017005 |

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

## Appendix 1

### Algorithm 1

Python pseudocode for reverse engineering proportional controller on each compute node

```python
def mpi_run_proportional_controller(combinations_subset):    gain = 0.2
# base gain    for ginmult, ginadd in combinations_subset:        stim_
time, stim_target = load_targets()        excitatory_input = load_base_
excitatory_input()
        for fit_iteration in range(0, 20):
            # setup simulation
             inhibitory_input = excitatory_input * ginmult + ginadd
             inhibitory_input = max(inhibitory_input, 1e-7)
        sim_data = mpi_run_neuron_sims(
            ginmult, ginadd, excitatory_input, inhibitory_input, stim_time)

        # process simulation result
        average_rate = get_average_rate(sim_data)
        mse = mean_squared_error(stim_target, average_rate)
        if mse < 0.5 and mn_recruited == 20:
          return inhibitory_input, mse, fit_iteration
        elif mse < 3:
            gain = 0.5 * gain
        elif mse < 1.5:
            gain = 0.25 * gain
         error = (stim_target - average_rate) * gain
    excitatory_input += error
    excitatory_input = boost_if_needed()
    excitatory_input = max(excitatory_input, 1e-7)
```

### Algorithm 2

Python pseudocode for simulating each motoneuron independently

```python
def mpi_run_motoneuron(tuple_motoneuron):
   motoneuron_h = load_neuron_data(tuple_motoneuron)
  while motoneuron_h.time_elapsed < 22000:
      motoneuron_h.step()
  return motoneuron_h.spike_times_soma
def mpi_run_neuron_sims(ginmult, ginadd, excitatory_input, inhibitory_
input):
  params = [(ginmult, ginadd, excitatory_input, inhibitory_input, mn_id)
            for mn_id in range(0, 20)]
  sim_data = mpi_parallel_context.map(mpi_run_motoneuron, params)
  return sim_data
```

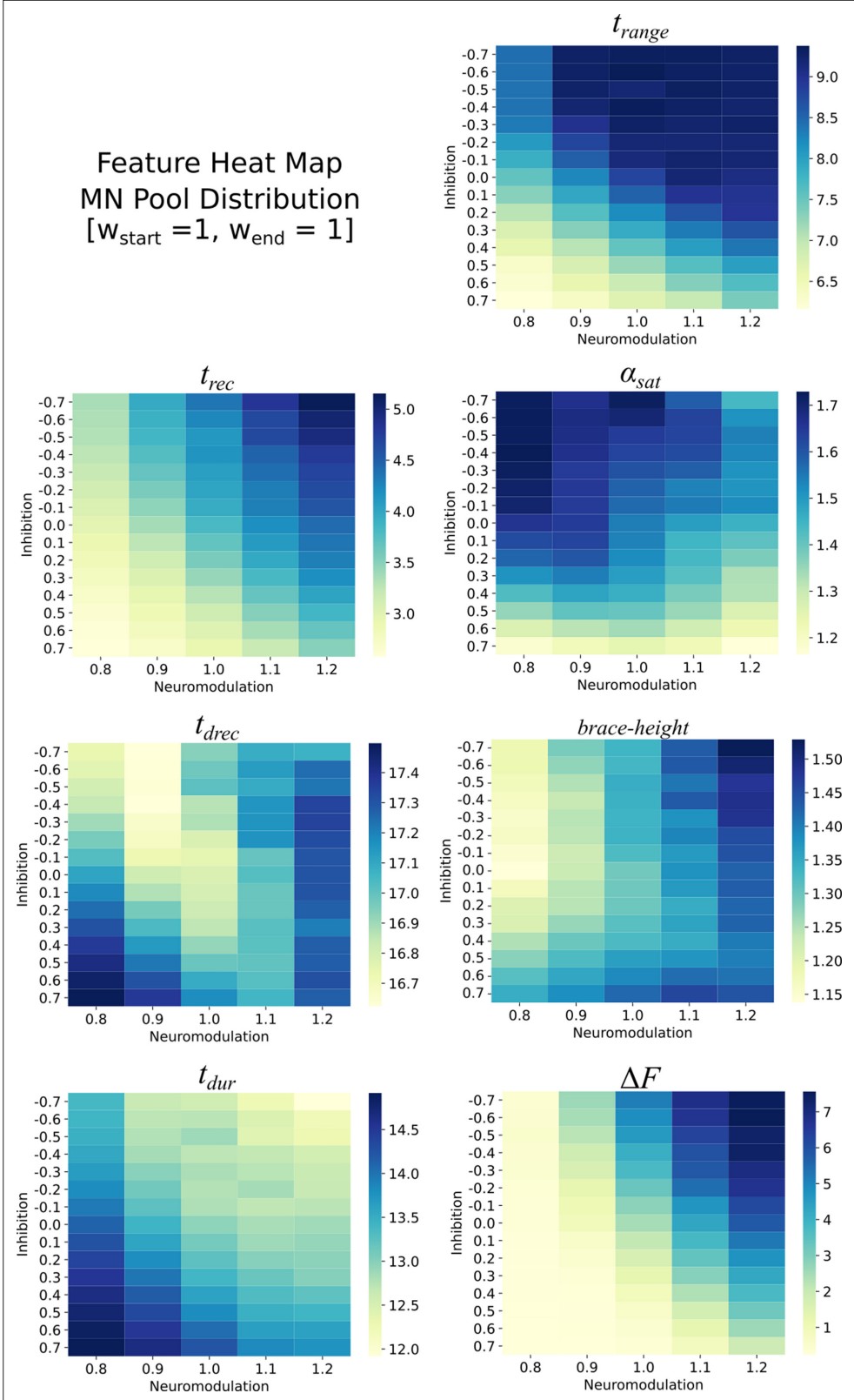

**Appendix 2—figure 1.** Heat maps for the different features extracted from the firing rates of the monotoneurons in the pool model. These heat maps are for the equally distributed excitatory weights . Each heat map is organized along the Neuromodulation and Inhibitory ranges tried in this work. The values within these heat maps show structure with respect to Neuromodulation and Inhibition and show that this structure can be used to reverse engineer the Neuromodulation and Inhibition themselves.

# Appendix 2

## Supplementary figures

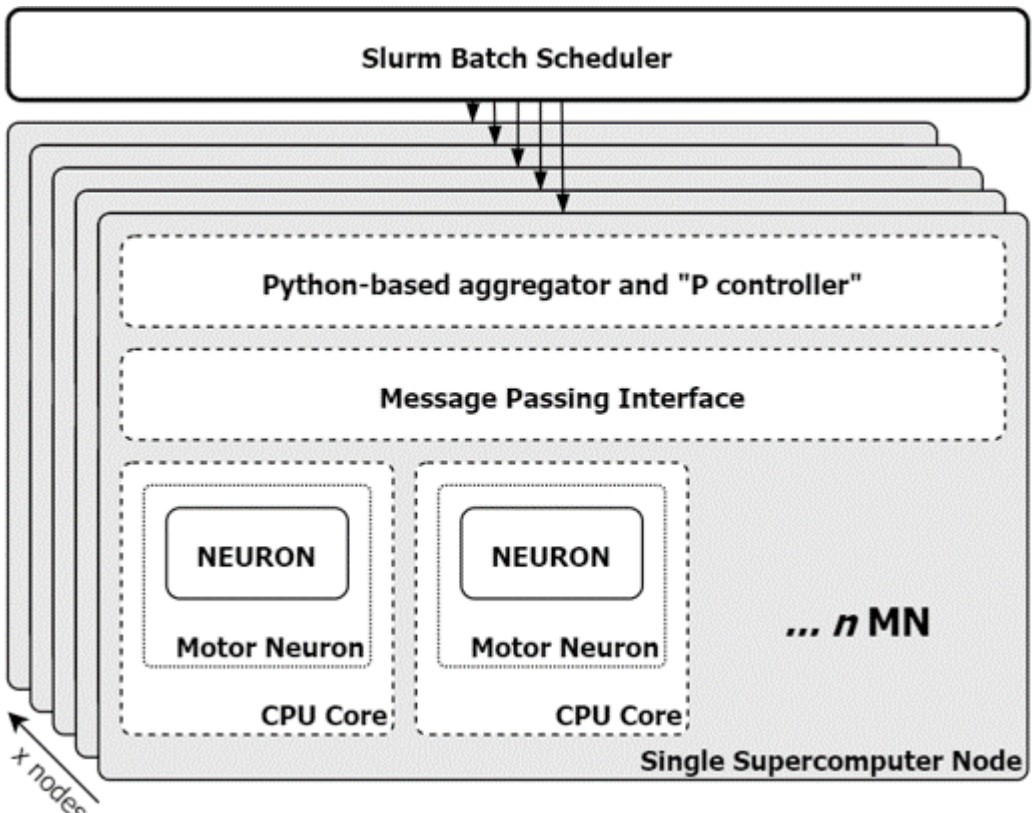

**Appendix 2—figure 2.** Schematic of the parallel architecture used to solve the motoneuron (MN) pool model and to find the solution for the input to that pool. Each neuromodulation, inhibition, excitatory input weight, and noise seed combination was assigned a node on the computer via the slurm batch scheduler. These 'hyper-parameters' were passed to each motoneuron which had a dedicated CPU core using the message passing interface (MPI). This work was done on the Bebop supercomputer at Argonne National Laboratory, Lemont, IL.

