## [Editor Report · eLife assessment]

The study by Chardon et al. is **fundamental** to advancing our understanding of presynaptic control of motor neuron output. Large-scale computer simulations were performed using well-established single motor neuron models to provide **compelling** evidence regarding the time-varying patterns of inputs that control motor neuron ensembles. The work will interest the community of motor control, motor unit physiology, neural engineering, and computational neuroscience.

---

## [Referee Report · Reviewer #1 (Public Review)]

The study presents an extensive computational approach to identify the motor neuron input from the characteristics of single motor neuron discharge patterns during a ramp up/down contraction. This reverse engineering approach is relevant due to limitations in our ability to estimate this input experimentally. Using well-established models of single motor neurons, a (very) large number of simulations were performed that allowed identification of this relation. In this way, the results enable researchers to measure motor neuron behavior and from those results determine the underlying neural input scheme. Overall, the results are very convincing and represent an important step forward in understanding the neural strategies for controlling movement.

---

## [Author Response]

The following is the authors’ response to the original reviews.

Reviewer 1“The exact levels of inhibition, excitation, and neuromodulatory inputs to neural networks are unknown.Therefore, the work is based on fine-tuned measures that are indirectly based on experimental results. However, obtaining such physiological information is challenging and currently impossible. From a computational perspective it is a challenge that in theory can be solved. Thus, although we have no ground-truth evidence, this framework can provide compelling evidence for all hypothesis testing research and potentially solve this physiological problem with the use of computers.”

Response: We agree with the reviewer. This work was intended to determine the feasibility of reverse engineering motor unit firing patterns, using neuron models with a high degree realism. Given the results support this feasibility, our model and technique will therefore serve to construct new hypotheses as well as testing them.

Common input structure lines 115I agree with the following concepts, but I would specify that there is not only one dominant common input. It has been shown that there are multiple common inputs to the same motor nuclei (e.g., the two inputs are orthogonal and are shared with a subset of the active motoneurons) particularly for agonist motoneuron pools of synergistic muscles. On the hand muscles the authors are correct that there is only one dominant common input. Moreover, there is also some animal work suggesting that common inputs is just an epiphenomenon. This is completely in contradiction to what we observe in-vivo in the firing patterns of motor units, but perhaps worth mentioning and discussing.

Response: Thanks for emphasizing this point. We have cited a recent reference discussing the important issue of common drive and the possibility of more than one source. Our simulations assume the net form of the excitatory input to all motoneurons in the pool is the same, except for noise. This net form (which produces the linear CST output in each case) essentially represents the sum of all inputs, both descending and sensory. Our results show the same over pattern as human data, i.e. that all motor unit firing patterns have similar trajectories (again allowing for the impact of noise). Future studies will consider separating excitatory inputs into different sources.

It is interesting that the authors mention suprathreshold rate modulation. Could the authors just discuss more on how the model would respond to a simulated suprathreshold current for all simulated motoneurons (i.e., like the ones generated during a suprathreshold-injected current or voluntary maximal feedforward movement?)

Response: Thank you for this point. Our use of the term “suprathreshold” was not applied correctly. We meant “suprathreshold” to refer to amount of input above the recruitment threshold. We have decided to remove this term so now the sentence “…so less is available for rate modulation…”.

194 a full point is missing.

Response: We addressed the error.

204-231 and 232-259, these two paragraphs have been copied twice.

Response: We addressed the error.

Line 475 typo

Response: We addressed the error.

591 It would be interesting to add the me it takes a standard computer with known specs and a super computer to run over one batch of simulation (i.e., how long one of the 6,300,000 simulation takes).

Response: Each simulation took about 20 minutes of real me. Assuming a standard computer with 16 processor cores using a similar microarchitecture as Bebop (Intel Broadwell architecture), the standard computer could run 16 simulations at a me (one simulation assigned per core). This would take the standard computer about 15 years to complete all 6.3M simulations.

594 I don't understand why there are 6M simulations, could the authors provide more info on the combinations and why there are 6M simulations.

Response: The 6M simulations are the total number of simulations that were performed for this work. A detailed explanation can be found in section: “Machine learning inference of motor pool characteristics” at line 591. Briefly, there were 315,000 simulations of a pool of 20 motoneurons (20 x 315,000 = 6.3 million). The 315,000 simulations was required to run all possible combinations of 15 patens of inhibition, 5 of neuromodulation, 7 of distribution of excitatory inputs and 30 different repeats of synaptic noise with different seeds. In addition, there were 20 iterations for each of these combinations to generate a linear CST output (as illustrated in Fig. 3). 15 x5 x 7 x 30 x 20=315,000.

In several simulations it seems that there was a lot of fine-tuning of inputs to match the measured motor unit firing pattern. Have the authors ever considered a fully black-box AI approach? If they think is interesting maybe it could spice up the discussion.

Response: We agree that AI has potential for reverse engineering the whole system and we are looking into adding it to future version of this algorithm as an alternative. We started with a simple but powerful grid search to enhance our understanding of the interaction between inputs, neuron properties and outputs.

**Reviewer 2**
Comment 1:“First, I believe that the relation between individual motor neuron behavioral characteristics (delta F, brace height etc.) and the motor neuron input properties can be illustrated more clearly. Although this is explained in the text, I believe that this is not optimally supported by figures. Figure 6 to some extent shows this, but figures 8 and 9 as well as Table 1 shows primarily the goodness of fit rather than the actual fit.”

Response: We agree with the reviewer that showing the relationship between the motor neuron behavioral characteristics (delta F, brace height etc.) and the motor neuron input properties would be a great addition to the manuscript. Because the regression models have multiple dimensions (7 inputs and 3 outputs) it is difficult to show the relationship in a static image. We thought it best to show the goodness of fit even though it is more abstract and less intuitive. We added a supplemental diagram to Figure 8 to show the structure of the reverse engineered model that was fit (see Figure 8D).

**Author response image 1. sa2fig1:** Figure 8. Residual plots showing the goodness of fit of the different predicted values: (A) Inhibition, (B) Neuromodulation and (C) excitatory Weight Rao. The summary plots are for the models showing highest 𝑅𝑅2 results in Table 1. The predicted values are calculated using the features extracted from the firing rates (see Figure 7, section Machine learning inference of motor pool characteristics and Regression using motoneuron outputs to predict input organization). Diagram (D) shows the multidimensionality of the RE models (see Model fits) which have 7 feature inputs (see Feature Extraction) predicting 3 outputs (Inhibition, Neuromodulation and Weight Rao).

Comment 2:“Second, I would have expected the discussion to have addressed specifically the question of which of the two primary schemes (pushpull, balanced) is the most prevalent. This is the main research question of the study, but it is to some degree le unanswered. Now that the authors have identified the relation between the characteristics of motor neuron behaviors (which has been reported in many previous studies), why not exploit this finding by summarizing the results of previous studies (at least a few representative ones) and discuss the most likely underlying input scheme? Is there a consistent trend towards one of the schemes, or are both strategies commonly used?”

Response: We agree with the reviewer that our discussion should have addressed which of the two primary schemes – push-pull or balanced – is the most prevalent. At first glance, the upper right of Figure 6 looks the most realistic when compared to real data. We thus would expect that the push-pull scheme to dominate for the given task.

We added a brief section (Push-Pull vs Balance Motor Command) in the discussion to address the reviewer’s comments. This section is not exhaustive but frames the debate using relevant literature. We are also now preparing to deploy these techniques on real data.

Comment 3:In addition, it seems striking to me that highly non-linear excitation profiles are necessary to obtain a linear CST ramp in many model configurations. Although somewhat speculative, one may expect that an approximately linear relation is desired for robust and intuitive motor control. It seems to me that humans generally have a good ability to accurately grade the magnitude of the motor output, which implies that either a non-linear relation has been learnt (complex task), or that the central nervous system can generally rely on a somewhat linear relation between the neural drive to the muscle and the output (simpler task).

Response: We agree with the reviewer, and we were surprised by these results. Our motoneuron pool is equipped with persistent inward currents (PICs) which are nonlinear. Therefore, for the motoneuron to produce a linear output the central nervous system would have to incorporate these nonlinearities into its commands.

Following this reasoning, it could be interesting to report also for which input scheme, the excitation profile is most linear. I understand that this is not the primary aim of the study, but it may be an interesting way to elaborate on the finding that in many cases non-linear excitation profiles were needed to produce the linear ramp.

This is a very interesting point. The most realistic firing patterns – with respect to human data – are found in the parameter regions in the upper right in Figure 6, which in fact produce the most nonlinear input (see push-pull pattern in Figure 4C). However, in future studies we hope to separate the total motor command illustrated here into descending and feedback commands. This may result in a more linear descending drive.